# Factors Associated with Unwanted Sexual Attention in Australian Nightlife Districts: An Exploratory Study of Nightlife Attendees

**DOI:** 10.3390/ijerph192316259

**Published:** 2022-12-05

**Authors:** Ryan Baldwin, Tavleen Dhinsa, Dominique de Andrade, Kerri Coomber, Ashlee Curtis, Samantha Wells, Jason Ferris, Cheneal Puljevic, Shannon Hyder, Steven Litherland, Peter G. Miller

**Affiliations:** 1School of Psychology, Deakin University, Geelong 3220, Australia; 2Department of Epidemiology and Biostatistics, Western University, London, ON N6A 3K7, Canada; 3Children’s Hospital of Eastern Ontario (CHEO) Research Institute, Ottawa, ON K1H 8LI, Canada; 4Better Outcomes Registry & Network (BORN), Ontario ON K1H 8L1, Canada; 5School of Psychology, The University of Queensland, Brisbane 4072, Australia; 6Institute for Mental Health Policy Research, Centre for Addiction and Mental Health (CAMH), Toronto, ON M6J 1H4, Canada; 7Dalla Lana School of Public Health, University of Toronto, Toronto, ON M5T 3M7, Canada; 8Department of Psychiatry, University of Toronto, Toronto, ON M5T 3M7, Canada; 9Centre for Health Services Research, The University of Queensland, Brisbane 4072, Australia; 10School of Public Health, The University of Queensland, Brisbane 4072, Australia

**Keywords:** unwanted sexual attention, childhood corporal punishment, masculine norms, trait aggression, nightlife, alcohol

## Abstract

Background: Experiences of unwanted sexual attention (UWSA) are commonplace within nightlife environments. While typically associated with aggression perpetration, literature has suggested that a history of childhood corporal punishment (CCP) may also be related to experiences of victimisation in nightlife environments. The current exploratory study aims to examine the associations between experiences of UWSA victimisation and a history of CCP, trait aggression, and conformity to masculine norms (Playboy and Winning), for males and females separately. Method: Street intercept interviews in the Brisbane inner-city entertainment precincts were used to measure demographic details and participants’ breath alcohol concentration. Online follow-up surveys were used to record participants’ experiences of UWSA on the night of interview, history of CCP, and self-reported rates of trait aggression and conformity to masculine norms. The final sample consisted of 288 females, as there were not sufficient male UWSA experiences for analysis. Results: Approximately 20% of female nightlife patrons experienced some form of UWSA victimisation. Logistic regression analyses identified that after controlling for age and intoxication, a history of CCP, trait aggression and masculine norm conformity were unrelated to experiences of UWSA for female respondents. Conclusions: The current study found that individual factors were unrelated to experiences of UWSA, indicating that simply being in the nightlife environment, especially as a female, increases the risk of UWSA victimisation. Understanding and exploring social and environmental risk factors, rather than individual factors, is needed to prevent victimisation in nightlife environments.

## 1. Introduction

Unwanted sexual attention (UWSA) refers to any non-consensual sexual touching or unsolicited sexual comments or physical gestures [1]. A growing body of research has provided evidence on the widespread occurrence of UWSA in late-night entertainment settings [2,3] and the potential factors influencing victimisation risk [4,5,6,7,8,9]. Within these environments, pervasive displays of aggression and overt sexual behaviours are often considered an accepted part of social behaviour that would not be tolerated in other public contexts [7]. These environments overwhelmingly place patrons, particularly young women, at an increased risk for sexual victimisation and persistent UWSA [7]. These experiences can have lasting impacts, increasing risk of distress, depression, anxiety or problematic alcohol use [10,11]. While evidence suggests that alcohol intoxication, masculinity, and sex expectations are associated with UWSA victimisation in the nightlife [3,6], less is known about whether experiences of childhood corporal punishment (CCP) are important underlying factors. While recent literature has identified that experiencing CCP increases such as the likelihood of experiences of verbal or physical aggression in the nightlife [12], it is still unclear whether this relationship is present for experiences of UWSA victimisation.

CCP is commonly defined as the non-injurious physical discipline of a child, used as a means of behaviour modification [13]. However, this practice is controversial, with 63 countries prohibiting its use within the home [14]. The distinction between CCP and physical abuse is argued to be one of legality rather than difference, due to their rate of co-occurrence [13,15] and similar adverse outcomes in adulthood, such as mental health disorders (e.g., depression and anxiety), risky substance use, and aggressiveness [16,17,18,19,20]. As such, it is believed that CCP is simply a part of the spectrum that is child abuse. Research suggests that child maltreatment or abuse may have an enduring impact on an individual’s self-esteem, increasing vulnerability to future victimisation [21]. This increased vulnerability has been found in studies showing elevated risk of physical and sexual revictimisation [21,22] as well as increased risk of experiencing intimate partner violence victimisation among those with a history of childhood maltreatment [23,24]. Research within this field has examined the relationship between childhood maltreatment or abuse and lifetime UWSA victimisation [21,22]; however, it is unclear whether this continuing vulnerability is present after experiencing CCP. Vulnerability of CCP appears to be present for verbal and physical victimisation when measured on a single night in the nightlife [12]. Only one cross-sectional study of university students in Finland was found on CCP and UWSA which showed a positive relationship between a history of CCP and experiencing UWSA in the nightlife environment [25]. The current exploratory study aims to examine whether this relationship is present for UWSA victimisation among young people attending nightlife settings in Brisbane, Australia.

While the risk of UWSA perpetration in the nightlife is related to personality factors such as masculine norm adherence and trait aggression [26,27], it is unclear whether these factors are similarly associated with UWSA victimisation. Laboratory-based research has demonstrated that females and males who do not conform to stereotypical gender roles are more likely to be targets of physical victimisation [28,29,30,31]. For example, Reidy et al. [29] found that hypermasculine men were more aggressive toward females who did not adhere to typically feminine gender role norms. As these findings are derived from laboratory-based aggression tasks, it is unclear whether this relationship would be present within the nightlife environment, where there is a high prevalence of UWSA [2,3]. Therefore, the current exploratory study aims to examine whether trait aggression proneness or tendency to act physically or verbally aggressive [32,33] and adherence or non-adherence to masculine norms of Winning (competitiveness) and Playboy (sexual promiscuity) [34] are related to UWSA victimisation in the nightlife. As previous research has highlighted a relationship between sexual assault victimisation and some feminine norms [35], we aimed to evaluate whether adherence to trait aggression and masculine norms, were negatively related to UWSA victimisation.

### The Present Study

The current exploratory study aims to explore the associations between experiences of unwanted sexual attention in nightlife environments and experiences of childhood corporal punishment, self-reported rates of trait aggression, and conformity to masculine norms (specifically Playboy and Winning), controlling for age and intoxication. As the influence of trait aggression and masculine norm adherence is expected to differ depending on the victim’s gender, the study aims to assess relationships for females and males separately.

It is hypothesised that: Unwanted sexual attention victimisation will be positively related to experiences of childhood corporal punishment.Unwanted sexual attention victimisation will be negatively related to trait aggression and masculine norm conformity.

## 2. Materials and Methods

### 2.1. Participants

The current study utilises participant data from a multisite project examining alcohol-related violence in patrons attending Queensland night-time entertainment districts [36]. Data were obtained via street intercept interviews in two inner-city suburbs of Brisbane, Australia (Fortitude Valley and West End) and a follow-up online survey. The total surveyed sample consisted of 288 females aged 18–43 (*M* = 22.96, *SD* = 5.81) and 202 males aged 18–50 (*M* = 23.11, *SD* = 6.03), all of whom completed the follow-up survey (see Table 1). Seventy participants (60 females and 10 males) experienced UWSA victimisation but no participants identified as instigating any UWSA. Due to the small number of males that experienced UWSA victimisation, all subsequent analysis is focused on female respondents. As such, the final sample for the current study consisted of 288 females aged 18–43 (*M* = 22.96, *SD* = 5.81).

### 2.2. Procedure 

Ethics approval was obtained from the Human Research Ethics Committees of Deakin University (2011-095) and The University of Queensland (2016001021). Study methods and descriptions of the full participant survey are explained in detail elsewhere [36]. Street intercept interviews occurred between 22:00 and 05:00 on Saturday nights in two Brisbane inner-city entertainment precincts—Fortitude Valley and West End—from March 2017 to November 2018. Small groups (*n* < 4) of trained researchers approached every third pedestrian in the night-time district to participate in a 10-min interview, asking about their demographics and their experiences leading up to their night out. Upon completion of the street interview, participants provided consent for their breath alcohol concentration (BrAC) to be recorded using Andatech Prodigy S breathalyser. Data was collected using Tap Forms^TM^ software on iPod Touches and iPhone devices. Participants provided their contact information and a follow-up survey link was sent to consenting participants the following day. The follow-up survey was active for up to seven days post interview and took approximately 15–20 min to complete. Of the 3010 participants that completed the on-the-night survey (approximately 74% of approached patrons), approximately 15% completed the follow-up survey. On-the-night interview responses were linked to online survey through the participants’ email address and phone number. However, this information was not stored with participant responses. While not compensated for their street interview, those that completed the follow-up survey received a AUD 20 gift card. The follow-up survey asked participants about their experiences of UWSA on the night of the interview, history of CCP, and levels of trait aggression and masculine norm conformity.

### 2.3. Measures

#### 2.3.1. On-the-Street Interview

##### Demographics

Due to limited responses from non-binary participants (*n* < 5), the current study modelled gender as a dichotomous variable. Due to the non-linear association often seen between age and alcohol use [37], the quadratic form of participant age was utilised in all analyses.

##### Time of Interview

The time of the street interview was recorded and coded as a continuous variable to indicate the hour in which the interview took place (i.e., 22:00 = 1, 23:00 = 2).

##### Breath Alcohol Concentration (BrAC)

Due to the established relationship between alcohol consumption and sexual violence in the night-time environment [3], alcohol was included as a covariate within the current study. BrAC at the time of interview was used to estimate participants’ alcohol intoxication levels. Based on previous literature [38], participants with a BrAC above 0.12 g/100 mL were coded as “intoxicated” and those with a BrAC at or below 0.12 g/100 mL were coded as “none to low intoxication”.

#### 2.3.2. Follow-Up Survey

##### Experiences of Unwanted Sexual Attention

During the follow-up online survey, participants were asked a range of questions including some specific to the night they were interviewed. Questions included “were you involved in any unwanted sexual attention (e.g., harassment, unwanted touching, sexual gestures)?” If participants responded “yes”, they were further prompted, “who was involved” and “who was the instigator” and selected from predetermined options (i.e., you, partner, close friend, acquaintance, stranger, security, other family, rival gang, police, other). 

##### Childhood Corporal Punishment

Experience of CCP was measured using a modified version of the Brief Physical Punishment Scale (BPPS) [39]. Participants identified on a five-point Likert scale ranging from 0 = “never”, 1 = “seldom”, 2 = “sometimes”, 3 = “often” and 4 = “very often”, the extent to which they experienced five types of physical punishment during their childhood. These were (1) pulled by the hair, (2) pulled by the ear, (3) slapped, (4) hit/kicked, and (5) hit with an object. The punishment described as “hit with the hand” in the original BPPS was modified to reflect two separate forms of punishment (“slapped” and “hit/kicked”). Participants that indicated “sometimes”, “often”, or “very often” to any of the BPPS questions were coded as experiencing CCP. The BPPS was found to have good internal reliability in the current study (Cronbach’s α = 0.86). 

##### Masculine Norms

Based on previous research examining alcohol-related aggression [40,41], two shortened subscales from the Conformity to Masculine Norms Inventory-46 (CMNI-46) [34] were utilised to measure participants’ adherence to Winning norms (competitiveness) and Playboy norms (sexual promiscuity) [34]. Three items from both scales were utilised to reduce survey length, which is related to participant attrition in young people in night-time setting [42]. Participants identified on a four-point Likert scale ranging from 1 = “strongly disagree”, 2 = “disagree”, 3 = “agree”, 4 = “strongly agree”, the extent to which the items related to them (i.e., “It’s important for me to win” or “If I could, I would frequently change sexual partners”). The CMNI-46 has been validated in both male and female samples to assess gender norm conformity [43]. Responses on both subscales were reversed where necessary and were averaged to determine adherence to masculine norms. Internal reliability of Winning and Playboy scales was acceptable in the current study (Winning α = 0.77; Playboy α = 0.76). 

##### Trait Aggression

Participants’ levels of trait aggression were measured using two items from the Trait Anger subscale of the Brief Aggression Questionnaire (AQ-12) [44]. The two items (“I have trouble controlling my temper”, and “I sometimes fly off the handle for no good reason”) were selected for inclusion based on their relationship with barroom aggression in previous analyses [45]. Using a five-point Likert scale, participants reported how characteristic the items were of themselves ranging from 1 = “Extremely uncharacteristic of me”, 2 = “Uncharacteristic of me”, 3 = “Neither uncharacteristic or characteristic of me”, 4 = “Characteristic of me”, and 5 = “Extremely characteristic of me”.

### 2.4. Analytic Plan

As the current study is examining experiencing of victimisation, only participants that did not identify as the instigator are included in analysis. Due to the small number of males that experienced UWSA victimisation (*n* = 10), the study did not possess sufficient power to adequately assess the influence of included variables on male UWSA victimisation. As such, all analyses pertaining to males has been included in Appendix A. First, we examined the inter-relations using Pearson and Point-Biserial correlations among the explanatory variables including age, trait aggression, adherence to Playboy and Winning, on the night BrAC, time of interview, and experiences of CCP. Next, we used logistic regression analysis to identify factors associated with UWSA. Logistic regression was chosen as it allows hierarchical model building to identify the impact of multiple predictors on a dichotomous outcome to be assessed, while also adjusting for multiple covariates. At step 1, we examined the associations between on the night experiences of UWSA and a history of CCP. At step 2, we introduced covariates (age, BrAC, and time of interview), and at step 3 we added personality and variables (trait aggression, adherence to Playboy and Winning norms). Model fit was examined through the Hosmer and Lemeshow goodness of fit test. Power analysis indicated that for females the current sample possessed sufficient power to find a medium effect (*OR* = 1.6, α = 0.05, power = 0.80). For males, the current samples only possessed sufficient power to find a very large effect (*OR* = 2.3, α = 0.05, power = 0.80). All analyses was conducted using SPSS 27 [46].

## 3. Results

### 3.1. Prelimiary Descriptives

Composite score means can be found in Table 1. Participants rated moderate to low across all psychosocial variables. Bivariate correlations for key variables can be found in Appendix A (see Table A1). Notably, over 50% of participants indicated they had some experiences of CCP, and 20% of the cohort experienced UWSA victimisation on the night of survey. Chi-square tests of association demonstrated no association between experiencing UWSA and experiencing CCP χ2 (1, N = 288) = 1.32, *p* = 0.25. All analyses relating to men are included in Appendix A (see Table A2, Table A3 and Table A4).

### 3.2. Correlates of Experience of Unwanted Sexual Attention

A hierarchical logistic regression model predicting UWSA was performed with the history of CCP entered at Step 1; age, BrAC, and hour interviewed entered at Step 2; and Playboy, Winning and trait aggression entered at Step 3. At Step 1, the model correctly identified 80.7% of cases, χ^2^ (1, N = 243) = 1.44 *p* = 0.23, while the final model (Table 2) correctly identified 81.1% of cases, χ^2^ (8, N = 243) = 13.30, *p* = 0.10. Regardless of the step, a history of CCP was not related to UWSA victimisation, and after controlling for all covariates, no psychosocial correlates were significantly related to experiencing UWSA in the nightlife environment. Included variables possessed low predictive ability in explaining experiences of UWSA (Nagelkerke R^2^ = 0.085).

## 4. Discussion

The current exploratory study aimed to examine associations between UWSA victimisation in nightlife entertainment districts and experiences of CCP, self-reported rates of trait aggression, and conformity to masculine norms (specifically Playboy and Winning), controlling for age, intoxication, and hour of interview. Contrary to hypotheses, experiences of CCP and the included psychosocial variables were not significantly related to experiencing UWSA in the nightlife environment. A notable finding, however, is that approximately 20% of female patrons, whereas only 5% of male patrons, reported experiencing UWSA on the night they were interviewed.

This study is the first to our knowledge that explores associations between historical experiences of CCP and UWSA victimisation on a single occasion in nightlife settings. Prior literature on this topic suggest childhood punishment impacts the victim’s self-esteem, which may in turn increase their future risk of sexual harassment [25]. Similar theories are shared in research examining the impact of childhood abuse on intimate partner victimisation, which suggests future victimisation stems from a learned acceptance or tolerance of abusive behaviour over time [24]. As experiences of nightlife UWSA are commonly perpetrated by individuals not known to the victim [47,48,49], and the current study focused on a single night, it is possible this longer term risk was not present. Rather, simply being present within the nightlife environment puts patrons at increased risk of UWSA, as they were likely exposed to external risk factors, such as overt displays of masculinity, peer dynamics, sex expectations and patron intoxication [8,50]. 

The present exploratory study found that trait aggression and the masculine norms of Winning and Playboy were also not associated with experiences of UWSA, indicating that those who violated traditional gender norms were not more likely to be victimised. As such, the current findings suggest that the measured personality variables are unrelated to sexual victimisation in the nightlife. Previous research has found a relationship between norm violation and victimisation; however, this latter research focused on physical victimisation, specifically in laboratory-based aggression tasks [29,30]. As such, it is likely that personality factors of the victim are unrelated to UWSA victimisation in the nightlife. Future research should explore the situational and social factors within the nightlife that place patrons at an increased risk of experiencing UWSA. This finding suggests that rather than focussing on characteristics of victims and what victims can do to prevent victimisation, prevention efforts would be more effective if they focus on perpetrators, venue management and environment to mitigate the risk of patrons experiencing UWSA [6].

Finally, we found that 20% of females in the current sample experienced UWSA on a single night out compared with only 5% of males. While this disparity may be related to stigma of males reporting sexual victimisation [51,52], this high proportion of victimisation in females reflects the large body of literature demonstrating females are at a greater risk of victimisation within the nightlife environment compared with males [3]. While there is limited evidence regarding effective UWSA prevention interventions [3], a growing body of literature supports the notion that victimisation may be reduced through managing environmental factors [53,54]. For example, research has indicated an association between venue factors such as alcohol promotion and availability, and staff monitoring with nightlife aggression [54,55,56]. Additionally, alcohol supply reduction policies such as outlet density and pricing have been associated with reductions in sexual crimes [57,58,59]. As such, there is a need for venues and governments to utilise these protective factors to introduce effective strategies and policies to reduce incidence of UWSA and mitigate patron harm.

### Limitations

The current study was unable to accurately assess whether alcohol consumption was related to UWSA, as BrAC was only recorded at the time of interview, but not at the time of victimisation. Victim intoxication or appearance of intoxication is associated with increased victimisation as perpetrators see the victim as more vulnerable or an easy target [3]. As it is unclear whether the experiences of UWSA occurred before or after the BrAC recording during street interview, or whether the individuals BrAC were increasing or decreasing, the current study was unable to examine this relationship. Future research examining this relationship should inquire around the approximate time of incident, or conduct matched exit interviews [60] to better understand how alcohol intoxication may interact with psychosocial and developmental risk factors in experiencing nightlife UWSA. Additionally, the current study was unable to conduct sex-based comparisons due to the low number of men who experienced UWSA. Future research should utilise substantially larger samples to ensure these relationships can be detected and analysed in male samples. Finally, the use of a subset of three items to measure each masculine norm factor in the current study may have impacted results, despite our internal reliability testing and prior research successfully using the same subset of items to measure Playboy and Winning norms [45].

## 5. Conclusions

Contrary to previous correlational evidence, the current exploratory study was unable to demonstrate associations between history of CCP, masculinity, trait aggression, and single night UWSA. Perhaps simply being in the nightlife environment, especially as a female, increases the risk of UWSA victimisation. The nightlife environment is one where UWSA is commonly experienced [5]. As such, venues need to ensure safety measures or that interventions are put in place to ensure their patrons can enjoy a safe environment in which they are not subject to victimisation. While evidence around interventions are still limited (Quigg et al., 2020), venues should work with local policy makers in implementing policies that reduce harms within the nightlight environment to help mitigate the likelihood of patrons experiencing UWSA victimisation.

## Figures and Tables

**Table 1 ijerph-19-16259-t001:** Sample Demographic Information for Female Respondents.

	Female (*n* = 288) ^a^
Trait Aggression	*M* = 1.81, *SD* = 0.89
Winning	*M* = 2.94, *SD* = 0.60
Playboy	*M* = 2.44, *SD* = 0.69
Age	*M* = 22.96, *SD* = 5.81
18–24 years	198 (68.8%)
25–34 years	53 (18.4%)
35–50 years	18 (6.3%)
Childhood Corporal Punishment(yes)	148 (52.7%)
BrAC ^b^ above 0.12 g/100 mL (yes)	51 (17.7%)
UWSA ^c^ Victimisation (yes)	60 (20.8%)

^a^ Represents valid responses. ^b^ BrAC = Breath Alcohol Concentration. ^c^ UWSA included as dichotomous variable with experience of UWSA reference category.

**Table 2 ijerph-19-16259-t002:** Hierarchical Logistic Regression of UWSA for Female Respondents.

	B	S.E.	Wald	*p*-Value	OR	95% C.I. OR		χ^2^	R^2^LL
Step 1								1.44	237.26
CCP ^a, b^	0.40	0.33	1.41	0.23	1.49	0.77	2.85		
Step 2								8.85	229.84
CCP	0.43	0.33	1.56	0.21	1.53	0.78	3.00		
Age	−0.08	0.05	2.26	0.13	0.93	0.84	1.02		
Age^2^	0.01	0.01	0.64	0.42	1.00	0.99	1.01		
BrAC ^c, d^	−0.56	0.40	2.00	0.16	0.57	0.26	1.2		
Hour Interviewed	0.13	0.12	1.12	0.27	1.14	0.90	1.5		
Step 3								13.30	225.39
CCP	0.30	0.36	0.70	0.41	1.34	0.70	2.70		
Age	−0.08	0.05	2.17	0.14	0.93	0.84	1.03		
Age^2^	0.01	0.01	0.70	0.40	1.00	0.99	1.01		
BrAC	−0.61	0.40	2.32	0.13	0.54	0.25	1.1		
Hour Interviewed	0.12	0.12	0.98	0.32	1.13	0.89	1.44		
Trait Aggression	0.22	0.19	1.39	0.24	1.25	0.86	1.81		
Winning	−0.24	0.29	0.73	0.40	0.78	0.45	1.37		
Playboy	0.40	0.24	2.63	0.11	1.49	0.92	2.40		

Hosmer and Lemeshow test indicated acceptable goodness of fit across all models (*p* > 0.05). ^a^ CCP = Childhood Corporal Punishment. ^b^ CCP included as dichotomous variable with no experience of Childhood Corporal Punishment reference category. ^c^ BrAC = Breath alcohol concentration. ^d^ BrAC included as dichotomous variable with BrAC < 0.120 reference category.

## Data Availability

Data sharing is not available due to ethics approval restrictions.

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
