# Peer review of "Factors Associated with Unwanted Sexual Attention in Australian Nightlife Districts: An Exploratory Study of Nightlife Attendees"

_ijerph, 2022, doi:10.3390/ijerph192316259_

Round 1

Reviewer 1 Report

The research approach has some advantages. Mention can be made of the originality of the topic, the location and timing of the street survey and the follow-up during the following days. At the same time, a number of disadvantages can be pointed out that lead to an overall negative evaluation. A model was presented that has not been confirmed in empirical studies. No independent variable was identified that was related to the main outcome (UWSA) in women and one in man (however not described) . This may also be due to the fact that other important factors were not considered or key variables were poorly defined.  The article also has a number of editing shortcomings, as if the authors did not read the IJERPH instruction at all.

Major comments

The objective and hypotheses are defined in a separate subsection. However, only hypothesis one o is clear, stated as positive, and the theoretical basis is explained in the introduction. With regard to hypothesis two, it is not clear whether the authors expect a positive or negative correlation. The introduction focuses on the positive correlation between being a perpetrator of UWSA and aggression trait or adherence to masculine norms.

These two factors (aggression trait  and adherence to masculine norms) were examined through questions in which the respondent describes himself or herself. It is unclear how masculine norms should be interpreted for women, for whom the results are discussed in more detail.  In the introduction there are references to research on the conforming to stereotypical gender norms.

The main analyzed, factor is UWSA defined on the basis of a series of experiences. Additional data were collected that were not described in the results and were not included in the analyses.

The main analyzed, factor is CCP defined on the basis of five different childhood experiences. 53% of women and 48% of men were classified as having such experiences.  This is an equal treatment of very frequent multiple experiences with sporadic ones. The authors should attempt alternative analyses, including, for example, defining three levels of CCP or a scale using complete information.

Tables with correlations add little and one may have reservations about the assumption of linear relationship,  use of Pearson's coefficient s well as including dichotomous variables. They could be moved to an supplementary appendix. The word reference category does not match the correlations in Tables 2 and 5.

Information about the percentage of UWSA appears in the abstract, tables and discussion, but it is not described in the results in the main text. The subsection on “UWSA prevalence” I missing. The quality of the study was greatly improved by a simple bivariate relationship between an otherwise defined CCP and UWSA (more than correlation). For example, the percentage of UWSA in the three CCP classes.

There is no information on the response rate. How many people refused to participate. How many of those participating in the street survey filled out a second online survey. Whether this follow-up disrupted anonymity. How the authors dealt with linking data?

Minor comments

Why are nightlife drinkers in the title of the article when about 80% of women and men did not meet the criterion for elevated BrAC.

Has an analogous paper on offenders been published?

Sample size should be in &2.1

Eliminating men is debatable. Regression models for 5% frequency are often considered. Only in the case of men any statistically significant result was obtained (relationship with the time of interview). 

The description of analysis methods should be supplemented with software.

Methods for assessing model fit not described in & 2.4 as well as the unexplained last column in Table 3. In the footnote under Table 3, it is worth noting which variables are dichotomous.

Are the means given in Tables 1 and 4 from the composite scales or as an average response per item.

In the limitations of the study, it is worth mentioning the use only selected questions from the primary tools cited as scales.

Although there is no significant variable, the model explains 8.5% of the variation in UWSA. Please check the recommended values for Nagelkerke's R-sq.

Editorial comments

Please, remove section 0 of which comes from the IJERPH template (from line 41).

Numbering of parts of the abstract is not necessary (lines 19,24,29).

There are errors in the numbering of subsections (3.1. twice).

The tables on men are moved to the appendix. If left that way, they should be numbered differently (table A1. A2…). There should be a reference to them in the main text with short comment.. However, I am in favor of including men and throwing the correlation tables outside the main text. Then tables 1 and 4 can be combined, and 3 and 6 will be in the main text.  

In Tables 2 and 5, please change the BrAC alignment in the first column (step 2).

References are cited in a different way than recommended with author names and year of publication, like APA rules.

Author Response

The author team thanks Reviewer 1 for their many helpful suggestions and detailed feedback, as addressed below. We hope that their inclusion improves the overall quality of the paper to be sufficient for publication.

Reviewer 2 Report

This study is extremely important. There are too many cases of victimization on the streets of cities in different countries and continents, regardless of the level of economic development or the political structure of countries. In addition, there is a clear lack of scientific knowledge about the causes of victimization (previously, personal factors were primarily seen). 

A few suggestions for the manuscript. Unfortunately, there is very little information about the sample. It would be desirable to put all the information about the sample in the paragraph "Participants". 

The regression analysis looks somewhat strange after the correlation analysis, since the latter did not reveal significant connections. Perhaps a simple analysis of variance should have been carried out, comparing two samples in all parameters – those with experience of victimization and those without experience of victimization, as well as those with experience of corporal punishment and those without this experience, which might have given a more obvious result.

It would be desirable to disclose table 3 in more detail, and to disclose abbreviations in the presentation of the results.  

It is not by chance that the authors in the article point out the importance of the environmental situation, the importance of the concentration of "bottling" establishments, but perhaps this is a complex problem, which is based on a certain behavior of the victim in certain circumstances. It is clear that in the office and during the day, the chance to experience of unwanted sexual attention is lower.

Author Response

The author team appreciate Reviewer 2 for providing feedback regarding interpretability and valuable insight into directions for future research.

Round 2

Reviewer 1 Report

Thank you very much for making thoughtful amendments and clarifying methodological issues 

Author Response

The author team appreciates your useful suggestions throughout this process.

Reviewer 2 Report

The team of researchers has finalized the presented material. However, it is necessary to justify the use of the regression method either in paragraph 2.4 or 3.2.

Author Response

Thank you for identifying further area for clarification. We have now provided justification for the logistic regression model in section 2.4 (Page 5. Line 214-217).

“Next, we used logistic regression analysis to identify factors associated UWSA. Logistic regression was chosen as it allows hierarchical model building to identify the impact of multiple predictors on a dichotomous outcome to be assessed, while also adjusting for multiple covariates.”